# Multimodal Prehabilitation in Patients Undergoing Complex Colorectal Surgery, Liver Resection, and Hyperthermic Intraperitoneal Chemotherapy (HIPEC): A Pilot Study on Feasibility and Potential Efficacy

**DOI:** 10.3390/cancers15061870

**Published:** 2023-03-20

**Authors:** Dieuwke Strijker, Wilhelmus J. H. J. Meijerink, Linda A. G. van Heusden-Schotalbers, Manon G. A. van den Berg, Monique J. M. D. van Asseldonk, Luuk D. Drager, Johannes H. W. de Wilt, Kees J. H. M. van Laarhoven, Baukje van den Heuvel

**Affiliations:** 1Department of Surgery, Radboud University Medical Centre, 6525 GA Nijmegen, The Netherlands; 2Department of Operating Rooms, Radboud University Medical Centre, 6525 GA Nijmegen, The Netherlands; 3Department of Rehabilitation, Radboud University Medical Centre, 6525 GA Nijmegen, The Netherlands; 4Department of Gastroenterology and Hepatology—Dietetics and Intestinal Failure, Radboud University Medical Centre, 6525 GA Nijmegen, The Netherlands

**Keywords:** colorectal neoplasm MeSH, neoplasm metastasis MeSH, hyperthermic intraperitoneal chemotherapy, colorectal surgery MeSH, postoperative complications MeSH, prehabilitation, perioperative care

## Abstract

**Simple Summary:**

Surgery plays a key role in the treatment of metastatic bowel cancer. Over the last years, more patients with metastatic bowel cancer are surgically treated, leading to increased survival rates. However, the surgical procedure is associated with a high risk of complications after surgery (up to 75%), such as bleeding, wound healing disorders, anastomotic leakage, and medical complications. Research has shown that prehabilitation improves outcomes after surgery for bowel cancer: it lowers the risk of complications and reduces the length of stay after surgery. Prehabilitation is a process of improving a patient’s condition between the time of diagnosis and the surgical procedure to enable a patient to withstand this stressful event. Most prehabilitation programs comprise multiple modalities, including an exercise program, nutritional intervention, psychological support, and intoxication cessation support. It is suggested that multimodal prehabilitation might also improve outcomes after procedures for metastatic bowel cancer. This pilot study aimed to determine the feasibility and potential efficacy of a prehabilitation program for patients undergoing surgery for metastases from bowel cancer.

**Abstract:**

Background: Surgery for complex primary and metastatic colorectal cancer (CRC), such as liver resection and hyperthermic intraperitoneal chemotherapy (HIPEC), in academic settings has led to improved survival but is associated with complications up to 75%. Prehabilitation has been shown to prevent complications in non-academic hospitals. This pilot study aimed to determine the feasibility and potential efficacy of a multimodal prehabilitation program in patients undergoing surgery in an academic hospital for complex primary and metastatic CRC. Methods: All patients awaiting complex colorectal surgery, liver resection, or HIPEC from July 2019 until January 2020 were considered potentially eligible. Feasibility was measured by accrual rate, completion rate, adherence to the program, satisfaction, and safety. To determine potential efficacy, postoperative outcomes were compared with a historical control group. Results: Sixteen out of twenty-five eligible patients (64%) commenced prehabilitation, and fourteen patients fully completed the intervention (88%). The adherence rate was 69%, as 11 patients completed >80% of prescribed supervised trainings. No adverse events occurred, and all patients expressed satisfaction with the program. The complication rate was significantly lower in the prehabilitation group (37.5%) than the control group (70.2%, *p* = 0.020). There was no difference in the type of complications. Conclusion: This pilot study illustrates that multimodal prehabilitation is feasible in the majority of patients undergoing complex colorectal cancer, liver resection, and HIPEC in an academic setting.

## 1. Introduction

Representing 10% of the global cancer incidence, colorectal cancer (CRC) is the third most common and second most deadly type of cancer worldwide [1]. Metastatic disease is the main cause of death in patients diagnosed with CRC. At time of diagnosis, up to 25% of patients already have metastases of the colorectal tumour, with hepatic, pulmonal, and peritoneal metastases as most common sites [2].

Both for the primary tumour and metastases in CRC, surgery is the basis in multimodal treatment regimens. Over the last years, more patients with metastatic disease are surgically treated, leading to increased 5-year overall survival rates for hepatic (approximately 50%) and peritoneal (approximately 30%) metastatic CRC [3,4].

Despite improved survival, high-risk procedures for complex primary and metastatic CRC, such as liver resection and hyperthermic intraperitoneal chemotherapy (HIPEC), in academic settings are associated with considerable morbidity, leading into lower health-related quality of life (QoL) and increased healthcare costs [5,6,7]. Global overall postoperative complication rates are approximately 30% after surgery for primary tumour resection and 35 and 75% after liver resection [7] and HIPEC, respectively [3,8].

Prehabilitation, a process to increase a patient’s functional capacity prior to a surgical procedure, has been shown to improve postoperative outcomes in primary CRC surgery in non-academic hospitals. Recent studies performed in non-academic hospitals have demonstrated reduced postoperative complications, fewer re-admissions, and shortened length of stay by multimodal prehabilitation in patients undergoing surgery for primary CRC [9,10,11]. Furthermore, a significantly higher percentage of prehabilitated patients are fully recovered in two months after surgery compared with non-prehabilitated patients (81 vs. 40%, respectively) [12].

These multimodal prehabilitation programs for primary CRC patients focus on targeting modifiable preoperative risk factors for morbidity after surgery, such as nutritional status, performance status, presence of anxiety and depression, and smoking [13]. Since these factors are also associated with poor outcomes after other types of major abdominal procedures, it is suggested that multimodal prehabilitation might also improve postoperative outcomes after procedures such as liver resection and HIPEC for metastatic CRC [14,15].

Studies on the effect of multimodal prehabilitation on outcomes in academic hospitals for complex primary and metastatic CRC patients have not yet been performed. Therefore, this pilot study aimed to determine the feasibility and potential efficacy of a multimodal prehabilitation program in patients undergoing complex colorectal surgery, liver resection, and HIPEC in an academic setting.

## 2. Materials and Methods

### 2.1. Design

This single-centre, historically controlled pilot study was conducted in the Radboud university medical centre in the Netherlands according to the ethical standards of the Helsinki Declaration (2013). The study started in May 2019 upon ethical approval of the Institutional Review Board and was completed in January 2020 after reaching a minimum of 15 patients in the intervention group [16].

### 2.2. Participants

All patients awaiting elective, curative surgery for complex primary CRC (colon or rectal resection), hepatic metastatic (liver resection), and peritoneal metastatic CRC (HIPEC) from May 2019 until January 2020 were considered potentially eligible for participation in the intervention group. Exclusion criteria for undergoing the program were: age < 16 years, expected surgery date < 3 weeks, paralysis or immobilization, American Society of Anesthesiologists (ASA) score ≥ 4, renal failure stage ≥ 3, and illiteracy.

Upon signed informed consent, patients were prospectively included in the trial and underwent a prehabilitation program. To determine the clinical effects of prehabilitation as a secondary outcome measure, the postoperative outcomes of the prospective cohort with prehabilitation were compared with a historical cohort without. These control patients were selected from patients undergoing surgery in the period from March 2017 until May 2019. An optimal matching method was used for (i) type of surgical procedure, (ii) age, (iii) ASA score, and (iv) current smoker yes or no with a ratio of 1:3.

### 2.3. Intervention

Eligible patients were offered to voluntarily participate in the prehabilitation program upon shared decision for surgery. Patients received both oral and written information on the content of the program and the study. After providing written informed consent, patients were scheduled for physiotherapeutic and dietitian baseline measurements within one week. The multimodal prehabilitation program included an exercise program, a nutritional intervention, psychological support, and smoking cessation support [17,18,19]. Appendix A provides detailed information on baseline measurements and the content of different modalities within the prehabilitation program.

The content of the exercise program was standardized and consisted of a high-intensity training (three times a week) and a low-intensity training (four times a week). The high-intensity training was supervised by a physiotherapist and adjusted based on a patient’s baseline measurements for the steep ramp test (estimated maximal oxygen consumption (VO2peak)) [20] and one repetition maximum (1RM) test [21]. The exercise program aimed to improve physical fitness (measured by VO2peak and 1RM) by at least 10%. For the low-intensity training, patients were instructed to aim for at least 60 min walking and/or cycling during the days without supervised training.

The goal of the nutritional intervention was to obtain an optimal nutritional status preoperatively. Therefore, a dietician provided personal advice on adequate energy and protein intake, with baseline measurements taken into account. Patients were also instructed to take daily protein (30 g whey protein, FrieslandCampina), multivitamin (50% of recommended dietary allowances), and vitamin D (10 μg for patients < 70 years and 20 μg for patients ≥ 70 years) supplements [22]. Following high-intensity training, patients took extra protein supplements.

Based on the Hospital Anxiety and Depression Scale (HADS) score [23], patients screened at risk (score of >15) were referred to psychological support to address anxieties, coping strategies, and postoperative expectations. Current smokers were referred to an outpatient smoking cessation program.

### 2.4. Perioperative Care

All patients included in this study underwent perioperative care following the Enhanced Recovery After Surgery (ERAS) program, which has been standardized care for the department of surgery for over 10 years [24]. The ERAS program includes patient education, general recommendations on perioperative nutrition and exercise, selected bowel preparation, multimodal analgesia, maintenance of perioperative normothermia, early oral intake and early mobilization, early removal of catheters and drains, and a pre-planned hospital stay according to clinical pathways. The surgical procedure was not delayed due to the prehabilitation program.

### 2.5. Primary Outcome Measure

The primary aim of this pilot study was to determine the feasibility of the multimodal prehabilitation program as measured by the following parameters:(1)Accrual rate: defined as the percentage of eligible patients who participated in the prehabilitation program. Reasons for non-participation were recorded.(2)Completion rate: defined as the percentage of patients who continued the prehabilitation program until surgery. Reasons for not continuing were recorded.(3)Adherence to the program: defined as the percentage of patients who continued prehabilitation until surgery and who completed at least 80% of the prescribed supervised trainings. Prescribed supervised trainings is calculated as the number of maximal trainings that could be performed in proportion to the duration of the preoperative period.(4)Satisfaction: assessed by a self-developed, specific questionnaire on patients’ satisfaction with the prehabilitation program.(5)Safety: defined as the number of adverse events occurring during the prehabilitation program.

### 2.6. Secondary Outcome Measure

The secondary endpoint of this pilot study was the efficacy potential of the prehabilitation program, measured by the effect on postoperative outcomes including 30-day complications using the Clavien–Dindo classification [25] and length of hospital stay in days. Postoperative outcomes were compared between the prospective intervention group and a historical control group.

### 2.7. Statistical Analysis

Because of the exploratory nature of this study, a formal sample size was not calculated. All data were analysed with IBM SPSS Statistics 25 (Armonk, New York, NY, USA). Data were assessed for normality using the Shapiro–Wilk test. Baseline demographics and clinical data were compared between intervention and control group using the independent samples *t*-test or chi-square test, as appropriate. To compare postoperative outcomes between both groups, an independent samples *t*-test or Mann–Whitney U test for continuous variables and chi-square test or Fisher’s exact test for categorical variables were conducted. A two-sided *p* value of <0.05 was defined as statistically significant.

## 3. Results

### 3.1. Feasibility Outcomes

From May 2019 until January 2020, 41 patients were screened for participation in this pilot study. Figure 1 shows the flow chart for inclusion and exclusion. Sixteen patients did not meet the inclusion criteria due to the surgical procedure planned within three weeks, leaving twenty-five patients eligible for inclusion.

Finally, sixteen patients commenced the prehabilitation program, resulting in an accrual rate of 64% (16/25 eligible patients). Two patients lacked motivation to undergo the prehabilitation program. Four patients had a change to the initial treatment plan shortly after being recruited and could therefore not start the prehabilitation program. Logistical problems withheld two patients from commencing prehabilitation in a timely manner. One patient was not able to perform physiotherapeutic baseline measurements due to physical limitations and therefore physical exercise was contraindicated. Table 1 summarizes the baseline characteristics of patients included in the intervention group.

### 3.2. Exercise Program 

In two cases, surgery was rescheduled within one week after onset of the prehabilitation program and subsequently the prehabilitation program was ended prematurely after completing one training session. The other 14 patients fully completed the intervention until the surgical procedure (completion rate 88%). The duration of the preoperative period, and thus the prehabilitation program, ranged from 7 to 46 days with a mean of 26 days. A total of 118 (average of 8.2 per patient) out of 131 (average of 9.2 per patient, =90%) prescribed training sessions were performed, with variability between patients of 50% to 133%. The adherence rate was 69%, as 11 patients completed >80% of the prescribed supervised trainings. Table 2 shows a detailed overview of feasibility outcomes.

### 3.3. Nutritional Intervention Program

All 16 patients underwent dietetic consultation with baseline measurements. Participating patients did not report any problems on taking prescribed supplements and reported to have taken both protein and vitamin suppletion according to prescriptions.

### 3.4. Psychological Support and Smoke Cessation Program

Based on the HADS score, one patient was screened as at risk and was referred for psychological support followed by one scheduled appointment. Three patients were instructed to start the smoking cessation program, of which two successfully stopped smoking preoperatively.

All patients expressed satisfaction with the prehabilitation program. Although some patients reported appointments linked to prehabilitation to be time-consuming, all patients felt empowered and supported by undergoing the program. No adverse events occurred during the study measurements or execution of the prehabilitation program.

### 3.5. Secondary Outcomes

For every patient in the intervention group (*n* = 16), three control patients were selected from a historical cohort (March 2017 until May 2019) using an optimal method for antecedently determined characteristics. For one patient, only two control patients were selected. Baseline characteristics of the 47 control patients are presented in Table 2 and were similar between the intervention and control groups.

Table 3 shows detailed information on secondary outcomes including postoperative complications and length of stay. The complication rate was significantly lower in the prehabilitation group (37.5%) as compared with the control group (70.2%, *p* = 0.020). Fewer severe complications occurred in the intervention group (12.5%) in comparison with the control group (36.2%) but the difference was not significant (*p* = 0.075). There were no differences in grade of complications between the two groups. The length of hospital stay was shorter in the prehabilitation group than the control group but did not differ significantly (median 6 days versus 9 days, respectively, *p* = 0.160).

## 4. Discussion

In this pilot study, a multimodal prehabilitation program was found to be safe and feasible in patients undergoing complex colorectal surgery, liver resection, and HIPEC, with an accrual rate of 64% and completion and adherence rates of 88% and 69%, respectively. The preliminary results of this study showed that undergoing prehabilitation improved postoperative outcomes including a lower risk of developing overall postoperative complications.

This is the first study on patients undergoing prehabilitation prior to complex colorectal surgery, liver resection, and HIPEC in an academic setting. Patients appeared to be motivated to participate in the prehabilitation program, with an accrual rate of 64%, which is higher than the only previous study evaluating prehabilitation before planned liver resection (accrual rate 33%) [27].

The principal reason for non-participation in this study seemed to be insufficient time between diagnosis and planned surgery (39%), rather than lack of motivation. This might be a logical consequence of the fact that no alternations were made in the preoperative workup and surgery planning since this was only a pilot study. Insufficient time for prehabilitation has been reported earlier as a barrier for prehabilitation in CRC surgery in non-academic hospitals [28], as implementation and execution of multimodal prehabilitation requires involvement and cooperation of many different healthcare providers. Although it is not desirable to adjust guidelines on surgery planning until the true effect of prehabilitation has been determined in complex primary and metastatic CRC surgery, future implementations of prehabilitation should acknowledge this barrier. The time between indication for a surgical procedure and actual surgery mostly depends on national agreements and differs subsequently among countries [29]. It is, however, assumed that delaying CRC surgery for a limited time does not harm patients or oncological outcomes [29]. Therefore, the benefits of prehabilitation should be weighed against the side effects of extending the preoperative period, such as psychological effects.

Once started with prehabilitation, 88% of patients fully completed the program. Two patients dropped out of the program due to rescheduling of the planned surgery with a resultant effect on the completion rate. Until scientific research fully supports the implementation of multimodal prehabilitation, it seems only understandable that operating room planning takes precedence over the prehabilitation program. As mentioned before, only adjustments in evidence-based guidelines for CRC care will support changes in this approach.

All previous studies on prehabilitation measured adherence rates based on the exercise program; however, heterogenous methods for determining adherence have been used. Subsequently, the numbers reported for adherence vary extensively [30,31]. However, one study on prehabilitation in patients awaiting liver resection reported a significantly higher adherence rate of 95% [27] compared with this pilot study. As the accrual rate in the study by Dunne et al. was low (33%), selection bias might have led to only including highly motivated and thus highly adhering patients, with a resultant impact on the protocol adherence. As compared with various feasibility studies on prehabilitation, the rate of patients remaining adherent to the prehabilitation protocol in this study was high (69%).

Overall, the high adherence rates of the different components of prehabilitation in this study (exercise program, nutritional intervention, and smoking cessation program) suggest that patients awaiting surgery for (metastatic) CRC in an academic setting are highly motivated for prehabilitation.

In line with previous prehabilitation studies in CRC surgery in non-academic hospitals [9,10,11], this study showed that prehabilitation improved postoperative outcomes in patients undergoing surgery for (metastatic) CRC in an academic setting. Whilst interpreting these results, it should be kept in mind that the primary objective of this pilot was not to determine the effect of prehabilitation on postoperative outcomes. Moreover, the risk for selection bias and attention bias should not be denied.

In this pilot study, it was decided to compare postoperative outcomes with a historic control group to determine the effect of prehabilitation. The results from this study show significant differences in complications, but time effects should be taken into consideration interpreting these results.

Although the overall complication rate of 70.2% in the historic control group corresponds with earlier reported numbers in patients undergoing HIPEC in the Netherlands [8], numbers on postoperative complication rates in surgery for primary and hepatic metastatic CRC are notably lower [3,7]. The high percentage of postoperative complications in the historical control group might be the result of the limited number of patients included in this study. Nonetheless, since the intervention group and the historic control group were matched for the most important independent preoperative risk factors for poor outcomes after abdominal surgery [14], it is plausible that the prehabilitation program had an important role in the considerable decrease of complications in this study.

In agreement with prior research on prehabilitation in CRC patients undergoing surgery in non-academic settings [11], this study showed that the postoperative hospital stay was shorter, although not significantly, in patients undergoing prehabilitation. Even though these results are encouraging, the reduction in length of stay cannot be fully attributed to the prehabilitation program. Guidelines concerning postoperative care are continuously updated and, subsequently, there are notable differences in the postoperative management between the intervention and the historic control cohorts. For example, it has always been routine to admit patients to the intensive care unit after a HIPEC procedure, but the time to transferring patients has been shortened over the years. The length of hospital stay has therefore already been decreased over time. However, this development and potential similar changes might not have led to the 3-day reduction in this study, making it likely that undergoing prehabilitation results in early discharge after surgery.

## 5. Conclusions

Surgery plays a key role in the treatment of complex primary CRC and metastatic CRC to improve survival rates but is associated with high morbidity rates. Following results of studies on enhancing recovery after primary CRC surgery in non-academic hospitals, prehabilitation might be a promising way to improve postoperative outcomes in patients undergoing high-risk procedures in academic settings. This pilot study illustrates that a multimodal prehabilitation program is feasible and safe for patients undergoing complex colorectal surgery, liver resection, and HIPEC for (metastatic) CRC. The preliminary positive effects of prehabilitation on postoperative complications in this study are encouraging for future studies to determine the true benefits of prehabilitation.

## Figures and Tables

**Figure 1 cancers-15-01870-f001:**
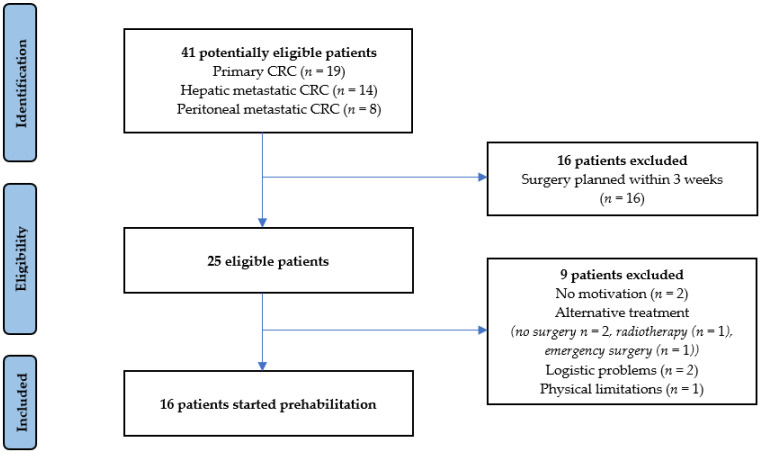
Flow chart for inclusion and exclusion.

**Table 1 cancers-15-01870-t001:** Baseline characteristics.

Characteristic	Prehabilitation Group (*n* = 16)	Control Group (*n* = 47)	*p*-Value
Age, mean years (SD)	67 (13.1)	68 (9.7)	*p* = 0.938
Male sex, *n* (%)	9 (56.3)	29 (61.7)	*p* = 0.772
Current smoker, *n* (%)	3 (18.8)	8 (17.0)	*p* = 0.875
Charlson comorbidity index (CCI) score, mean score (SD)	7.6 (2.0)	7.6 (1.8)	*p* = 0.988
Neoadjuvant therapy, *n* (%)			*p* = 0.064
Chemotherapy	7 (43.8)	7 (14.9)	
Radiotherapy	0	1 (2.1)	
Chemoradiation	2 (12.5)	3 (6.4)	
ASA score, *n* (%)			*p* = 0.941
II	8 (50)	24 (51.1)	
III	8 (50)	23 (48.9)	
Surgical procedure for, *n* (%)			*p* = 0.998
Primary CRC	9 (56.2)	26 (55.3)	
Left hemicolectomy	2 (12.5)	6 (12.8)	
Right hemicolectomy	2 (12.5)	6 (12.8)	
APR	5 (31.2)	14 (29.8)	
Hepatic metastatic CRC	4 (25.0)	12 (25.5)	
Segmental resection	3 (18.8)	9 (19.1)	
Hemihepatectomy	1 (6.2)	3 (6.4)	
IMM classification			
Grade I	1 (6.3)	4 (8.5)	
Grade II	2 (12.5)	5 (10.6)	
Grade III	1 (6.3)	3 (6.4)	
Peritoneal metastatic CRC			
Single organ resection +	3 (18.8)	9 (19.1)	
HIPEC	0	0	
Multiple organ resection +	3 (18.8)	9 (19.1)	
HIPEC			
CCR score			
0	3 (18.8)	9 (19.1)	
1	0	0	
2	0	0	
3	0	0	
Surgical technique, *n* (%)			*p* = 0.978
Open	14 (87.5)	41 (87.2)	
Laparoscopic	2 (12.5)	6 (12.8)	

*n* = number of patients; SD = standard deviation; ASA = American Society of Anesthesiologists; CRC = colorectal carcinoma; APR = abdominal perineal resection; IMM = Institut Mutualiste Montsouris complexity classification [26]; CCR = completeness of cytoreduction.

**Table 2 cancers-15-01870-t002:** Feasibility outcomes.

Accrual Rate, *n* (%)	16/25 (64%)
Completion rate, *n* (%)	14/16 (88%)
Duration of prehabilitation program, mean days (SD; min–max)	26.2 (10.7; 7–46)
Prescribed supervised training sessions, mean (SD; min–max)	9.2 (4.7; 1–18)
Completed supervised training sessions, mean (SD; min–max)	8.2 (4.1; 1–14)
Adherence rate exercise program, *n* (%)	11/16 (69%)
Adherence rate nutritional intervention, *n* of patients with reported protein and vitamin suppletion according to prescriptions (%)	16/16 (100%)
Adherence rate smoke cessation program, *n* of current smokers with successful smoking cessation prior to surgery (%)	2/3 (67%)

*n* = number of patients, SD = standard deviation, min = minimum, max = maximum.

**Table 3 cancers-15-01870-t003:** Secondary outcomes.

Postoperative Outcome	Prehabilitation Group (*n* = 16)	Control Group (*n* = 47)	*p*-Value
Overall postoperative complication rate, *n* (%)	6 (37.5)	33 (70.2)	*p* = 0.020 *
Clavien–Dindo Classification, grade #, n (%)			
I	3 (18.8)	7 (14.9)	
II	1 (6.3)	9 (19.1)	
IIIa	1 (6.3)	6 (12.8)	
IIIb	0	2 (4.3)	
IVa	1 (6.3)	3 (6.4)	
IVb	0	3 (6.4)	
V	0	3 (6.4)	*p* = 0.342 *
Severe postoperative complications (Clavien–Dindo > IIIa) rate, *n* (%)	2 (12.5)	17 (36.2)	*p* = 0.075 *
Length of stay, median days (IQR)	6 (5–9)	9 (5–14)	*p* = 0.160 ^$^

*n* = number of patients; IQR = interquartile range; # some patients had multiple postoperative complications: the complication with the highest Clavien–Dindo grade is given; * chi-squared test; ^$^ Mann–Whitney U test.

## Data Availability

The data that support the findings of this study are available from the corresponding author D.S. upon reasonable request.

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
