# Peer review of "Multimodal Prehabilitation in Patients Undergoing Complex Colorectal Surgery, Liver Resection, and Hyperthermic Intraperitoneal Chemotherapy (HIPEC): A Pilot Study on Feasibility and Potential Efficacy"

_cancers, 2023, doi:10.3390/cancers15061870_

Round 1
Reviewer 1 Report
This study investigates feasibility of multimodal prehabilitation in patients with complex primary and metastatic cancer awaiting complex colorectal surgery, liver resection or HIPEC in an academic hospital. Of the 25 eligible patients, 16 started the prehabilitation of whom 14 completed the intervention. 11 patients completed >80% of the prescribed trainings. Compared to the control group, the prehabilitation group had lower complication rates (70.2% vs 37.5%) and less Clavien-Dindo grad II complications (40.4% vs 12.5%). Therefore, multimodal prehabilitation seems feasible.
It is a well written article. However, it has a major flaw in the statistical analyses. Clavin-Dindo should not be tested by Mann Whitney U test but by a Chi-square test. It does not make sense to compare class II Clavin Dindo in both groups and then state a difference. Please adjust this result throughout the article.
Line 78: Health-related quality of life is an outcome measure, mentioned in reference 11. The authors in this particular article mention coping of anxiety. You could consider mentioning health-related quality of life as an outcome in line 80.
Line 127. Which measurement was used to measure the 10% increase in fitness?
Lines 160-165: how was the efficacy of the nutritional intervention, anxiety training, and smoking cessation defined/assessed?
Table 3. The Clavin-Dindo classification should not be compared per class but by a Chi-squared test.
Shouldn’t the highest complication Clavin Dindo count?
Therefore it is not appropriate to mention only the significant difference in class II of Clavin Dindo. Please do the proper analyse and rephrase this result throughout the article.
Conclusions (lines 354-357). These lines are not a conclusion of this article and can therefore be omitted.
Line 459: It is not clear whether 30 g protein are supplemented in total or a total of 60 g is supplemented.
Reviewer 2 Report
I would like to thank Dr. Strijker and co-authors for their manuscript submission titled “Multimodal prehabilitation in patients undergoing complex colorectal surgery, liver resection and hyperthermic intraperitoneal chemotherapy (HIPEC): a pilot study on feasibility and potential efficacy” to Cancers. Included in the reviewed materials, in addition to the manuscript, are one (1) figure and three (3) tables. I appreciate the opportunity to review their submission and please accept my appraisal after a thorough evaluation.
Synopsis
The authors present the results of a pilot study on the feasibility of prehabilitation in patients undergoing complex colorectal surgery, liver resection and HIPEC. In their primary measures, they achieved 64% accrual, 88% completion, 69% adherence with 100% patient satisfaction and no adverse events. As a secondary measure, prehabilitation was associated with an overall decrease in postoperative complications as compared to a matched historical cohort.
Critique
As a proof of concept, the manuscript functions to demonstrate a reasonable compliance with prehabilitation. I am not confident the secondary conclusion that it is associated with a decrease in postoperative complications is appropriate given the difficulty with case matching in the historical cohort however I appreciate the authors have presented that limitation in their discussion. To promote the primary strength of the publication, namely the compliance and adherence rates, I have asked for further clarification. With revision, I feel the paper would be acceptable for publication.
1) Regarding the completion and/or adherence to prehabilitation, would the study authors be able to provide what the rates were for each individual element (i.e. exercise, nutritional, psychological, smoking cessation, etc.)?
2) Could the details of the surgical procedures be expanded upon? What do the authors define as “complex” colorectal surgery? Do the liver resections span major hepatectomy to non-anatomic wedge? What were the results of the cytoreduction in patients undergoing HIPEC? Were procedures all completed for curative intent?
Minor criticisms
- page 4, line 110. Insert numeral for four i.e. “…score, and (iv) current smoker…”
Reviewer 3 Report
The authors have written a paper on the feasibility and potential efficacy of a multimodal prehabilitation program in patients undergoing surgery in an academic hospital for complex primary and metastatic CRC.
This is an untreated topic, but it is always interesting to understand the role of prehabilitation especially in oncological patients who undergo surgery.
But I still have a few comments to make.
1. The authors should explain the type of liver surgery the patients underwent,
2. The degree of complexity of liver surgery is related to complications and length of hospitalization, I recommend citing this article “Conditional cumulative incidence of postoperative complications stratified by complexity classification for laparoscopic liver resection: Optimization of in-hospital observation”, https://pubmed.ncbi.nlm.nih.gov/36041926/ to enhance the discussion.
3. the type of neoadjuvant therapy if any
4. ERAS programs often improve pulmonary performance through exercise. Authors should discuss about the pulmonary complications, how prehabilitation could prevent these events.
Round 2
Reviewer 3 Report
No more comments. The authors improved the manuscript.